# Co-Inoculation Between Bacteria and Algae from Biological Soil Crusts and Their Effects on the Growth of *Poa annua* and Sandy Soils Quality

**DOI:** 10.3390/microorganisms13081778

**Published:** 2025-07-30

**Authors:** Lin Peng, Xuqiang Xie, Man Chen, Fengjie Qiao, Xingyu Liu, Yutong Zhao, Xiawei Peng, Fangchun Liu

**Affiliations:** 1School of Biological Sciences and Technology, Beijing Forestry University, Beijing 100083, China; pl32784775@163.com (L.P.); xuqiangxie@163.com (X.X.); yuanfei1000@163.com (M.C.); qiaofengjiezuiqian@163.com (F.Q.); xingyu66@bjfu.edu.cn (X.L.); dustinuniverse@126.com (Y.Z.); 2Shandong Academy of Forestry, Jinan 250014, China

**Keywords:** biological soil crusts, bacteria–algal co-inoculation, aggregate stability, growth promotion, SQI

## Abstract

Microorganisms (bacteria and algae) are important components of biological soil crusts, which exhibit crucial functions in promoting plant growth, maintaining soil structure, and improving soil nutrient content. To determine the effects of combined inoculation on the growth of *Poa annua* and sandy soils, four species of bacteria and algae were isolated and identified from biological soil crusts (during different developmental stages in a karst rocky desertification area). The soil quality was evaluated based on a soil quality index (SQI), growth indicators of *Poa annua*, soil physicochemical properties, and a stability analysis of aggregates. With the application of nutrient-poor sandy soils as the substrate, different treatment inoculation solutions were inoculated onto *Poa annua*. The results revealed that bacteria–algal co-inoculation reduces soil acidity, enhances soil nutrient content and aggregate stability, improves soil quality, and protects plant growth. Notably, compared with the single application of bacterial solution and algal solution, the combined application of bacteria–algal solution significantly improves the sandy soil quality.

## 1. Introduction

Soil nutrients serve as the basis of soil fertility, which is a comprehensive reflection of the physical and chemical properties of soil. Functionally, soil nutrients serve as the basic source of nutrients required for plant growth, and they are directly connected with the level of soil or land productivity. Therefore, the enhancement of soil nutrients is crucial for the improvement of soil fertility. Some existing studies on soil fertility improvement are based on the addition of microbial agents, such as the research of Zhang et al. [1], in which it was found that the microbial quantity and nutrient content in soil (treated with microbial agent CAMP) were significantly improved, and the CFU was 357.1 times that of CK. Moreover, the contents of TN, TP, AN, and AP in sandy soils were 4.3–10.1%, 7.9–10.0%, 7.6–24.1%, and 3.7–8.4% higher than those in CK, respectively.

In addition, some relevant studies on soil fertility improvement were based on the addition of a mixture of microorganisms and microalgae. For example, Zhao et al. [2] found that compared with CK treatment, microalgae combined with *Bacillus subtilis* significantly increased the ammonium nitrogen, nitrate nitrogen, DOC, and pH in soil by 11.93%, 17.80%, 5.58%, and 2.75% on the 35th day of culture, respectively. However, the above methods exhibit limited effects in improving soil fertility, and research on soil structure is relatively lacking, which underscores the importance of further studies.

Biological soil crusts (BSCs) are complex surface coverings, which are formed by algae, lichens, mosses, and microorganisms bonding with soil surface particles through mycelium, rhizoid, and secretions [3]. As an important part of biological soil crusts, the microbial community exhibits crucial functions in maintaining ecosystem stability, promoting nutrient circulation, alongside the assistance of the establishment and sustainable survival of vascular plants, etc. [4,5]. With the significant changes in the community composition of the succeeding microorganisms in biological soil crusts, the abundance of algae exhibits a constant increasing trend [6]. In 1995, Zhou et al. [7]. initially investigated the desert crusts in the early dry area of Shapotou, Zhongwei, Ningxia, and 13 genera were identified in this study, which were composed of 5 genera of cyanobacteria, 6 genera of green algae, and 2 genera of diatoms. Notably, soil crusts tend to grow in harsh environmental conditions, which can indicate that the organisms in soil crusts exhibit strong adaptability to the harsh environment with relatively poor nutrition.

Notably, there are interactions in chemical exchange, cell signal transduction, and gene transfer between bacteria and algae in soil crusts [8]. Existing studies predominantly focus on the practical application of the combination of bacteria and algae [9] (such as artificial cultivation of biological crusts and soil amendment in saline-alkali [10,11]), while the relevant research on the effects of the interaction between bacteria and algae on plant growth and poor soil quality is relatively insufficient. Based on the above findings, the application of microorganisms and algae from biological soil crusts in plant growth is investigated in this study in order to provide a theoretical basis for the ecological restoration of the infertile land.

*Poa annua*, a resilient species exhibiting remarkable stress tolerance and soil stabilization capacity, demonstrates significant potential in rehabilitating degraded ecosystems. Capitalizing on its ecological adaptability, this study innovatively integrates *Poa annua* with functionally optimized microorganisms isolated from biological soil crusts (BSCs) to establish a synergistic plant–microbe remediation framework. A tri-factorial experimental design—featuring functional bacterial suspensions, algae suspensions, and bacterial–algae suspensions—was implemented, with precision rhizosphere inoculation ensuring targeted microbial colonization. We systematically investigated the regulatory effects of microbial treatments on (i) physio-ecological traits of *Poa annua*, (ii) soil physicochemical properties, and (iii) aggregate stability dynamics. A soil quality index (SQI) model was employed for multidimensional quantitative assessment. This work presents a cost-effective and ecologically sustainable plant–microbe co-remediation paradigm, offering both theoretical insights and practical methodologies for restoring nutrient-depleted soils.

## 2. Materials and Methods

### 2.1. Test Materials

The experiment was started on 14 September 2023 in Saning Garden Nursery Test Base of Beijing Forestry University. The soil was obtained from Daxing District of Beijing, and the basic physicochemical properties of the tested sandy soils were as follows: pH 8.20, conductivity 60.24 μs·cm^−1^, organic carbon 11.31 mg·kg^−1^, alkali-hydrolyzed nitrogen 143.33 mg·kg^−1^, available phosphorus 32.2 mg·kg^−1^, total nitrogen 0.25 g·kg^−1^, total phosphorus 0.13 g·kg^−1^, available potassium 22.4 g·kg^−1^, and polysaccharide content 0.5157 mg·kg^−1^.

The biological soil crusts were collected from the karst desertification area of Jianshui County, Honghe Hani and Yi Autonomous Prefecture, Yunnan Province. This area has no obvious ground vegetation, with an altitude of 1500 m, and belongs to a typical South Asian tropical monsoon climate region (Appendix A).

### 2.2. Isolation of Functional Bacteria

Functional microorganisms were isolated and identified through a comprehensive screening approach combining high-throughput screening techniques, pot culture assays, and functional characterization methods (methodological details in Appendix A).

During the pot experiments in this section, the cultivation results of the water control group and medium control group were basically the same. The components of the culture medium in the algal suspension were simpler than those in the bacterial suspension, and no difference could be found versus the results of the water control group. Given that the watering method is one of the most common methods in the outdoor application (with the advantages of simple operation and low-cost measure), water was applied in the control group for the subsequent experiments, and the influence of the culture medium components on the soil and plants was relatively negligible.

### 2.3. Isolation of Algae

The target algal strains were isolated through shake-flask cultivation and repeated purification procedures, with their growth rates being comparatively evaluated based on colony development on agar plates (methodological details in Appendix A).

### 2.4. Preparation of the Inoculation Solutions

The 4 functional bacterial strains obtained by the above screening were inoculated into 10% TSB liquid medium and cultured in a shaking incubator (30 °C, 180 rpm) for 36 to 48 h. The concentration of the bacterial solution was adjusted to 10^8^ CFU/mL with distilled water, followed by mixing in equal volume ratios to prepare bacterial solution A. Similarly, the 4 algal strains were inoculated into BG11 liquid medium and cultured with 2700 lx of light at 27 °C for 4 weeks. The concentration of the algal solution was adjusted to 10^6^ CFU/mL with distilled water, followed by mixing in equal volume ratios to prepare algal solution Z. Finally, bacterial solution A and algal solution Z were mixed in equal volume ratios to prepare bacterial–algal solution AZ.

### 2.5. Design of Experiment

A flowerpot with an outer diameter of 10.5 cm, a height of 8 cm, and an inner diameter of 8 cm was selected for the subsequent experiment, and a water leakage hole existed at the bottom of the flowerpot. Each flowerpot was filled with 500 g of sandy soil (with 100 full *Poa annua* seeds inside). The A, Z, and AZ treatments were applied in liquid form, with 80 mL of solution in each pot. Notably, the appropriate amount of application could prevent the liquid from flowing out from the bottom leakage hole in order to avoid bacterial loss. Simultaneously, an equal volume of water control (CK) was set, with 5 replicates in each group. The bacteria–algal solution was applied once every 7 days, and the substrate was sprayed with water each day during the rest time to keep it wet (culture cycle was 90 days). The seeds of *Poa annua* were obtained from the School of Soil and Water Conservation, Beijing Forestry University.

### 2.6. Soil Physicochemical Measurements

Measurements of soil properties (methodological details in Appendix A) [12] included pH, soil conductivity (EC), soil organic carbon (SOC), alkali hydrolyzed nitrogen (AN), available phosphorus (AP), total nitrogen (TN), total phosphorus (TP), available potassium (AK), soil polysaccharide, structure and stability of the soil aggregate.

### 2.7. Microbial Biomass Measurements

Measurements of microbial biomass (methodological details in Appendix A) included microbial biomass carbon (MBC) and microbial biomass nitrogen (MBN).

### 2.8. Poa annua Growth and Physicochemical Measurements

A precision electronic autobalance (XPR205/AC, Mettler, Greifensee, Switzerland) was employed to measure the total weight, aboveground biomass, and underground biomass of plants, while the total length, plant height, and root length of plants were measured directly by ruler, and the nitrogen and phosphorus contents of plant leaves were determined by a continuous flow analyzer (AA3, SEAL, Frankfurt am Main, Germany).

### 2.9. Calculation of Soil Composite Index

The comprehensive evaluation of soil quality can be generally divided into three steps: factor selection, weight determination, and comprehensive index acquisition. Due to the continuous changes in the nature of soil factors, the membership function of continuous quality is adopted for each evaluation index [13], while the fluctuation of the membership function distribution is determined by the positive and negative values of the principal component factor load. The ascending distribution function and descending distribution function are calculated as follows [14]:*Q*(*x*_*i*_) = (*X*_*ij*_ − *X*_*imin*_)/(*X*_*imax*_ − *X*_*imin*_)(1)*Q*(*x*_*i*_) = (*X*_*imax*_ − *X*_*i*_)/(*X*_*imax*_ − *X*_*imin*_)(2)
where *Q*(*x_i_*) represents the membership value of each soil factor, *X_ij_* represents the measured value of each soil evaluation index, and *X_i_* max and *X_i_* min represent the maximum and minimum values of the measured values of item I soil quality evaluation index.

Given the common differences in the status and importance of each factor of soil quality, the weight coefficient tends to be employed to express the importance of each factor [15]. In this study, principal components were employed in SPSS software to investigate the load of each index in order to obtain the variance contribution rate and cumulative contribution rate of the principal components of each factor, alongside the calculation of their weights. The calculation formula is as follows [16]:

(3)$$W_{i}=/C_{i}\sum_{i=1}^{n}Ci$$where *W_i_* represents the weight value of the *i*th soil index in a certain principal component; *C_i_* represents the absolute value of the load of the *i*th soil index in a certain principal component; and n represents the number of evaluation indicators.

According to the additive and multiplicative rule, the index value of each soil quality is synthesized by multiplication, and *SQI* under different treatments is calculated as follows [17]:(4)$$\mathrm{SQI}=\;\sum_{i=1}^{n}W_{i}*\;Q(x_{i})$$
where *W_i_* is the weight value of soil index I; *Q*(*x_i_*) is the membership value of each soil factor. The indices including pH, EC, SOC, AN, AP, TN, TP, SWC, C/N, C/P, N/P, AK, EPS, MBC, MBN, MBC/MBN, R0.25, MWD and GMD were used to calculate SQI, and the corresponding comprehensive indexes were the quality of soil nutrients and physical structure and the activity of soil microorganisms.

### 2.10. Statistical Analysis

Excel 2016 and Origin 2022 were employed for data processing and mapping, SPSS 24.0 statistical software was used for analysis of variance (ANOVA), and the LSD (*p* < 0.05) method was employed in the difference significance analysis.

## 3. Results

### 3.1. Impact of Bacteria–Algae Co-Inoculation on Physicochemical Properties of Sandy Soils

According to Table 1, all treatments (Z, A, AZ) significantly increased soil pH compared to the control (CK), with increments of 0.04, 0.07, and 0.16, respectively, demonstrating that the AZ treatment effectively ameliorated soil acidity. Correspondingly, electrical conductivity (EC) exhibited a remarkable enhancement of 53.35% after the AZ treatment (*p* < 0.05). Moreover, both Z and AZ treatments elevated soil moisture content by 12.4% and 11.3%, respectively. The organic carbon (SOC) contents in CK, Z, A, and AZ treatments were 7.09, 8.09, 9.44, and 15.23 g·kg^−1^, respectively. Compared to CK, SOC increased by 14.1%, 33.15%, and 114.8%, while total nitrogen (TN) rose by 57.69%, 65.38%, and 80.77%, and total phosphorus (TP) increased by 161.54%, 138.46%, and 161.54%, significantly altering the soil C:N:P stoichiometric ratios (*p* < 0.05). Notably, these treatments significantly enhanced the soil available nutrients. The alkali-hydrolyzable nitrogen (AN) followed the order AZ > A > Z > CK (*p* < 0.05). Compared to the CK groups, Z, A, and AZ treatments significantly boosted AN, AP), and AK (*p* < 0.05), and the most pronounced enhancement could be found in AZ treatment—79.54%, 50.03%, and 46.36%, respectively. Additionally, soil polysaccharide content was significantly elevated across all treatments (*p* < 0.05). These findings indicate that the application of the synthetic algal–bacterial consortium substantially improved the availability and overall fertility of soil nutrients.

### 3.2. Impact of Bacteria–Algal Co-Inoculation on Composition and Stability of Sandy Soils Aggregates

Generally, higher values of aggregate stability indices (MWD, GMD, and R_0.25_) coupled with lower values of PAD, E_LT_, and *K* indicate improved soil aggregation and structural stability. As illustrated in Figure 1, the dominant fraction across all treatments was micro-aggregates (<0.25 mm). However, compared to the control (CK), treatments Z, A, and AZ significantly (*p* < 0.05) promoted the formation of macro-aggregates (>2 mm and 1–2 mm fractions). Notably, the 1–2 mm fraction increased by 93.2%, 83.02%, and 170.53%, respectively, under Z, A, and AZ treatments, while the 0.25–1 mm fraction decreased significantly (*p* < 0.05). In addition, the AZ treatment induced a significant reduction (*p* < 0.05) in the proportion of water-stable aggregates <0.25 mm. These results demonstrate that AZ inoculation differentially promoted the formation of medium-to-large aggregates (>1 mm), with particularly pronounced effects on macroaggregates (>2 mm). Furthermore, AZ application significantly enhanced key aggregate stability indices including MWD, GMD and R_0.25_ while simultaneously reducing PAD, E_LT_, and K values (Table 2). Compared to CK controls, all treatments showed marked improvements in aggregate stability, with MWD increasing by 21.31–62.3% (*p* < 0.05) and GMD elevating by 22.73–45.45% (*p* < 0.05). The AZ treatment emerged as the most effective strategy for enhancing soil aggregate stability, demonstrating synergistic effects surpassing individual microbial applications.

### 3.3. Impact of Bacteria–Algal Co-Inoculation on Microbial Biomass

Compared to the control (CK), all treatments significantly enhanced both microbial biomass carbon (MBC) and nitrogen (MBN) contents (*p* < 0.05; Figure 2). Specifically Z treatment increased MBC by 10.2% and MBN by 20.09%, A treatment elevated MBC by 17.28% and MBN by 47.14%,. The AZ consortium showed the most pronounced stimulation, boosting MBC by 34.28% and MBN by 70.19%. Concurrently, the MBC/MBN ratio decreased significantly (*p* < 0.05) across all treatments: Z’s 8.23% reduction, A’s 20.06% reduction, and AZ’s 21.97% reduction. The AZ co-inoculation demonstrated significant synergistic effects on microbial biomass carbon (MBC) and nitrogen (MBN) (*p* < 0.05).

### 3.4. Impact of Soil Physicochemical Properties on Aggregate Stability and Microbial Biomass

To further elucidate the regulatory effects of soil physicochemical properties on aggregate stability and microbial biomass, Mantel tests were employed to analyze the correlations between key environmental factors and soil structural stability (Figure 3). The results demonstrated highly significant positive relationships (*p* < 0.01) between aggregate stability and soil organic carbon (SOC), total nitrogen (TN), available nitrogen (AN), available potassium (AK), and extracellular polysaccharides (EPSs), suggesting their potential roles in enhancing soil aggregation through either promoting organo-mineral complex formation or stimulating microbial secretion of binding agents. Furthermore, microbial biomass showed significant correlations (*p* < 0.01) not only with these factors but also with electrical conductivity (EC), available phosphorus (AP), and the nitrogen-to-phosphorus ratio (N/P), indicating that microbial growth and metabolism are simultaneously regulated by both salinity conditions and nutrient stoichiometry. These findings collectively demonstrate that soil nutrient contents and their stoichiometric characteristics influence aggregate formation and stabilization through dual pathways: directly by providing cementing substances and indirectly by modulating microbial activity.

### 3.5. Impact of Bacteria–Alga Co-Inoculation on the Growth of Poa annua

Compared with the control (Table 3), the total length, root length, plant height, total weight, leaf total nitrogen, and total phosphorus contents of all treatment groups were significantly increased (*p* < 0.05), and the effects of the combined inoculation group were superior to that of the single inoculation group. The total length of treatment groups increased by 95.3% to 184.7%, root length by 105.0% to 190.0%, plant height by 86.7% to 180.0%, total nitrogen content in leaves by 26.5% to 113.1%, and total phosphorus content in leaves by 413.0% to 656.5%. Overall, all treatments significantly increased the underground and above-ground biomass, and the above-ground biomass of the AZ group was significantly different from the control (*p* < 0.05).

### 3.6. Comprehensive Index of Soil Quality Under the Combined Inoculation of Bacteria and Algae

Radar diagrams provide an intuitive visualization of both individual soil parameter status and overall soil quality, where greater distance from the origin indicates better parameter performance and larger enclosed areas reflect superior overall conditions. The radar plot of average membership degrees for different treatments (Figure 4) revealed that in the CK treatment, organic carbon showed the lowest average membership (0.363) while C/P ratio exhibited the highest (0.653). For the Z treatment, available phosphorus had the minimum membership (0.337) whereas mean weight diameter (MWD) demonstrated the maximum (0.807). In the A treatment, C/P ratio recorded the lowest membership (0.343) while total nitrogen (TN) showed the highest (0.659). The AZ treatment displayed minimum membership in MBC/MBN ratio (0.34) and maximum in R0.25 (0.688). Parameters with the lowest membership values represent the primary limiting factors for soil quality assessment, while those with highest values reflect treatment-specific improvement effects. Our analysis demonstrates that the Z treatment primarily enhanced soil aggregate stability, bacterial inoculant (A) mainly improved total nitrogen content, and their combination (AZ) predominantly increased the mass fraction of >0.25 mm aggregates.

Principal component analysis (PCA) of 10 soil quality indicators revealed three principal components (PCs) with eigenvalues ≥1, collectively accounting for 100% of the total variance, with the first three PCs capturing the majority of information from the original 19 indicators (Table 4). The first principal component (PC1), explaining 68.99% of the variance, showed high loadings (≥0.9) for soil pH, SOC, AN, TN, AK, MBC, MBN, MWD, GMD, and R0.25, with respective weights of 0.973, 0.922, 0.952, 0.954, 0.905, 0.986, 0.97, 0.966, 0.947, and 0.92, indicating strong intercorrelations among these parameters and demonstrating that soil nitrogen content, microbial biomass, and aggregate stability are key determinants of overall soil quality. PC2 accounted for 24.02% of variance, with high loadings for C/N (0.9) and C/P (0.948) ratios, highlighting the importance of carbon-nitrogen-phosphorus stoichiometry in soil quality assessment. PC3 (7% variance explained) was primarily weighted by soil water content (0.844). These findings demonstrate that the evaluated soil quality indicators interact dynamically, collectively influencing soil ecological conditions through their interdependent relationships.

The soil quality index (SQI) was calculated by incorporating the indicator weights into Equation (4), with the results presented in Figure 5. Significant differences in SQI values were observed among treatments, following the order AZ > A > Z > CK. The bacterial–algae co-inoculation (AZ) achieved the highest SQI value of 0.52885, demonstrating its superior efficacy in substantially enhancing overall soil quality compared to individual microbial treatments.

The weights of each index were substituted into Equation (4) to calculate SQI (Figure 5). According to the results, different treatments also exhibited significant differences in the soil quality comprehensive index, and the order of SQI was AZ > A > Z > CK. The SQI index of the mixed treatment of AZ was 0.52885, followed by the synthetic bacterial solution treatment (A), and the improvement effects of synthetic algal solution treatment (Z) were similar to that of synthetic algal solution treatment (CK), which were both significantly higher than the control.

## 4. Discussion

### 4.1. Influencing Mechanism of Bacteria–Algal Co-Inoculation on Soil Physicochemical Properties

According to the results, the pH of the soil in the bacterial–algal solution treatment group exhibited a certain increasing trend. Once algae colonized and proliferated in the soil, they would consume CO_2_ (in water and air) through photosynthesis, and they could absorb CO_2_ in bicarbonate and transform the bicarbonate into carbonate [18]. On the other hand, algae could secrete alkaline extracellular substances, thus alkalizing the surrounding environment [19]. Therefore, the pH level could be increased based on these two effects, which also indicates that the synthetic bacteria–algal solution might exhibit a significant auxiliary function in the pH recovery of acidic soil [20]. As shown in Table 1, the treatment of bacterial algal solution increased the content of soil nutrient elements to a certain extent, which might further increase the EC value [21]. Notably, soil organic carbon serves as an important source of N, P, K, and other nutrient elements in soil, while the available N, P, and K in soil are nutrient elements that can be absorbed and utilized by plants. It has been revealed by many studies that the application of microbial fertilizer could enhance the carbon and nitrogen metabolism of soil, alongside the promotion of soil nutrient cycling and improvement of soil nutrient utilization efficiency [22,23]. In this research, the parameters (including the contents of soil organic carbon, alkali-hydrolyzed nitrogen, available phosphorus, total nitrogen, total phosphorus, and available potassium) all exhibited an increasing trend after the application of synthetic bacterial solution, alongside the significant improvement of soil nutrient status (*p* < 0.05). Generally, the sandy soil is relatively poor with a low nutrient content, while the growth of algae may provide the soil with other sources of nitrogen, such as nitrate nitrogen [24]. Algal cells would secrete exopolysaccharides during growth, which could further improve soil fertility and promote the growth of microorganisms [25]. Besides, the soil carbon, nitrogen, phosphorus, and polysaccharides were significantly correlated with the stability of soil aggregates and microbial biomass, and these substances exhibited positive effects on the stability of aggregates and microbial biological activity after the application of the mixed solution. Overall, it could be concluded that microorganisms can increase soil nutrient content by secreting polysaccharides and other cementing substances, which could further improve the stability of aggregates, weaken the restriction of microbial nutrients, and promote microbial metabolic activity.

### 4.2. Influencing Mechanism of Bacteria–Algal Co-Inoculation on the Formation and Stability of Aggregates

Soil structure serves as an important factor for soil fertility, alongside the migration and transformation of substances in soil. Soil particles tend to interact with each other and aggregate to form aggregates of different sizes, which rarely exist in the form of single grains. Notably, the aggregate with an appropriate structure can promote the transformation of soil nutrients [26,27]. It was found in this study that the content of medium and large-sized aggregates (>2 mm and 1~2 mm) was increased after the application of bacteria and algae mixture, while the content of micro-aggregates (<0.25 mm) exhibited a decreasing trend, which was similar to the results of other researchers [28,29]. The above phenomenon could be attributed to the exopolysaccharides secreted by the bacteria alga solution, which can improve the water holding capacity of soil, cement, and agglomerate soil [30,31], thus further enhancing the stability of soil structure [32,33]. Furthermore, the application of the mixture of bacteria and algae was confirmed to enhance the soil microbial biomass and microbial activity, thus producing more organic secretions and promoting the mutual agglomeration of soil particles [34,35]. Notably, the structure failure rate, unstable aggregate index and K were inversely proportional to the stability of aggregates, while R0.25, MWD, and GMD were proportional to the stability of aggregates [36,37]. It has been indicated by numerous studies that the organic matter exhibits a good cementation effect on soil water-stable aggregates [38,39], and its rich humic acid and other substances can cement mineral soil particles together, or form soil water-stable aggregates through the action of metal ion bonds in the soil [40,41,42].

### 4.3. Influencing Mechanism of Bacteria–Algal Co-Inoculation on Microbial Biomass

The effects of microbial fertilizer on the improvement of soil microbial activity exhibit a gradually enhancing trend, which could be attributed to the adaptation of microbial fertilizer to the environment. With the continuous propagation of effective bacteria, the soil water condition was continuously improved, alongside the gradual improvement of the soil environment [43,44]. Functionally, microbial biomass can reflect the strength of microbial activity, the addition of bacteria and algae mixture can improve soil water content, and create favorable conditions for microbial growth. Besides, the addition of bacteria and algae mixture could further increase soil microbial biomass carbon and nitrogen [45,46], improve soil physical structure, and promote soil organic carbon decomposition and soil nutrient cycling [47]. Generally, R-strategies with rapid growth and turnover are dominated by microorganisms, which tend to exhibit a low level of MBC/MBN. On the contrary, K-game microorganisms with slower growth and turnover rate generally have higher MBC/MBN [48,49]. This study also showed that the nitrogen addition significantly reduced MBC/MBN, which could further verify the ability of the combined application to accelerate the turnover of microorganisms and increase the nitrogen content [50].

### 4.4. Promoting Effect of Bacteria–Algal Co-Inoculation on Poa annua, Growth

It was confirmed by studies that the inoculation of synthetic flora could promote the growth of above-ground and underground parts of plants, alongside the increase of their biomass [50]. In this study, different inoculation groups significantly increased aboveground and underground biomass, root length, plant height and leaf nitrogen and phosphorus contents. Notably, the AZ treatment exhibited the most significant effects on TP and TN, alongside the root length of bluegrass. It could be hypothesized that the functional superposition and functional complementarity between synthetic bacteria and algae could improve their phosphorus solubilization and nitrogen fixation capabilities, which contribute to the effective absorption of nitrogen and phosphorus for plants in soil. Overall, this synergistic effect could increase the plant height and root length, alongside the further improvement of plant biological yield.

### 4.5. Comprehensive Evaluation of Soil Quality Under the Bacteria–Algal Co-Inoculation

In this study, the comprehensive score of soil quality for the combined inoculation treatment of AZ was the highest, which could confirm that the combined inoculation could take full advantage of the synergistic effects of bacteria and algae [51]. During the process of growth and metabolism, algae could secrete various organic compounds and active substances (such as sugars, amino acids, peptidesm and hormones) for bacterial growth, while the metabolites of bacteria could be utilized by algae. For example, nitrates produced by bacteria could be used as nutrients for algae growth, while carbon dioxide could be used by algae to synthesize organic matter [52]. Based on certain interactions (such as chemical exchange, cell signaling and gene transfer) between bacteria and algae, the application of bacteria–algae symbiosis is superior to that of a single microbe [8,53]. Compared with the single inoculation, the combination of bacteria and algae exhibited better performance to promote soil carbon mineralization, increase microbial populations, reduce the use of fertilizer, enhance the crop root density, and increase the crop indoleacetic acid production [54]. Moreover, bacterial and algal symbiosis could promote plant growth, improve seed germination rate, enhance plant indole acetic acid production and the activity of multiple defense enzymes [55]. Based on the joint application of rhizosphere growth-promoting bacteria and cyanobacteria, the use of nitrogen fertilizer could be significantly reduced [56], while mutually-beneficial symbiosis between microorganisms could further enhance the respective capabilities.

## 5. Conclusions

This study pioneers a paradigm shift from conventional single-species inoculation approaches by isolating multiple high-efficiency bacteria and algae from biological soil crusts across karst rocky desertification succession gradients, developing a novel bacteria–algae consortium with demonstrated synergistic effect, and employing multi-dimensional (plant–soil–microbe) analysis to elucidate the “nutrient enhancement–structural improvement–microbial activation” tripartite mechanism driving plant growth. Our results demonstrate that the co-inoculation between bacteria and algae significantly enhanced soil aggregation, microbial carbon and nitrogen content, as well as plant growth parameters. This study establishes a novel conceptual framework for ecological restoration. Our research achieved valuable results and significance in controlled experiments, but the stability and adaptability of the bacterial–algal combination in actual complex environments still need in-depth study.

## 6. Patents

Peng Xia-Wei, Chen Man, SUN Xiao-Yu et al. A strain of Streptomyces fusiformis LS159 and its application: CN202211465909.5[P]. 2023-02-03.Zhou Jinxing, Chen Man, GUAN Yinghui et al. A new soil improvement microbial agent, preparation method and application: CN202210974181.2[P]. 2022-12-16.

## Figures and Tables

**Figure 1 microorganisms-13-01778-f001:**
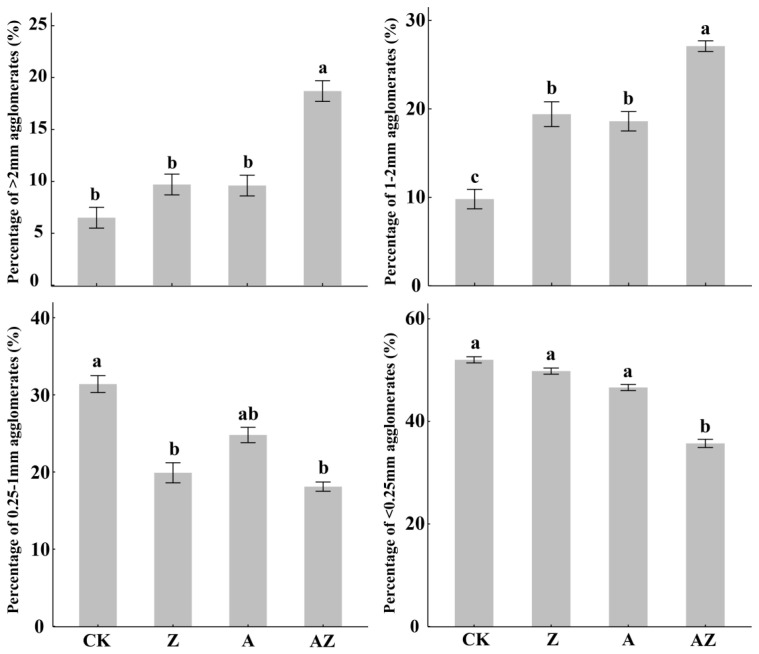
Percentages of soil aggregates with various sizes under different treatments. The vertical lines represent the standard deviation, while the different lowercase letters indicate significant differences between treatments (*p* < 0.05).

**Figure 2 microorganisms-13-01778-f002:**
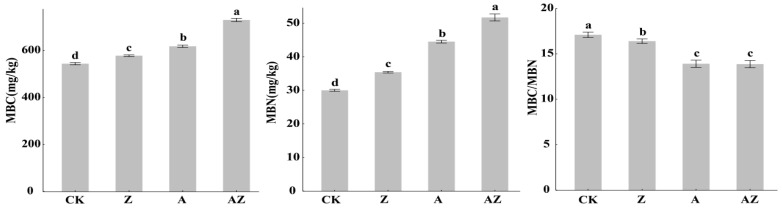
The effects of different treatments on MBC, MBN, and MBC/MBN. The vertical lines represent the standard deviation, and different lowercase letters indicate the significant differences between treatments (*p* < 0.05). MBC: microbial biomass carbon, MBN: microbial biomass nitrogen.

**Figure 3 microorganisms-13-01778-f003:**
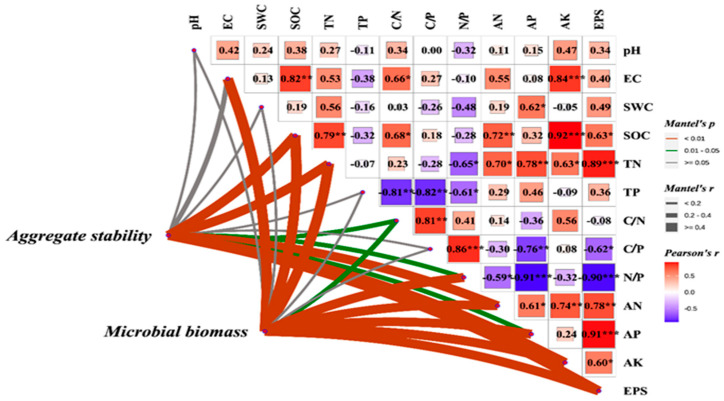
Effects of soil physicochemical factors on aggregate stability and microbial biomass. SOC: soil organic carbon, EPS: exopolysaccharides, TN: total nitrogen, AN: available nitrogen, AK: available potassium, AP: available phosphorus. (*: *p* < 0.5, **: *p* < 0.05, ***: *p* < 0.01).

**Figure 4 microorganisms-13-01778-f004:**
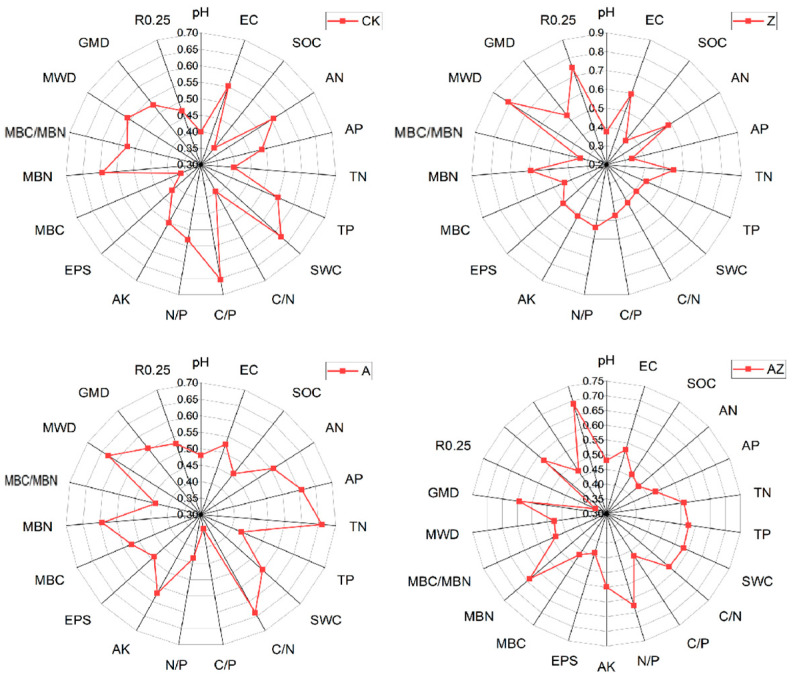
Radar plot of the average membership degree of each soil indicator.

**Figure 5 microorganisms-13-01778-f005:**
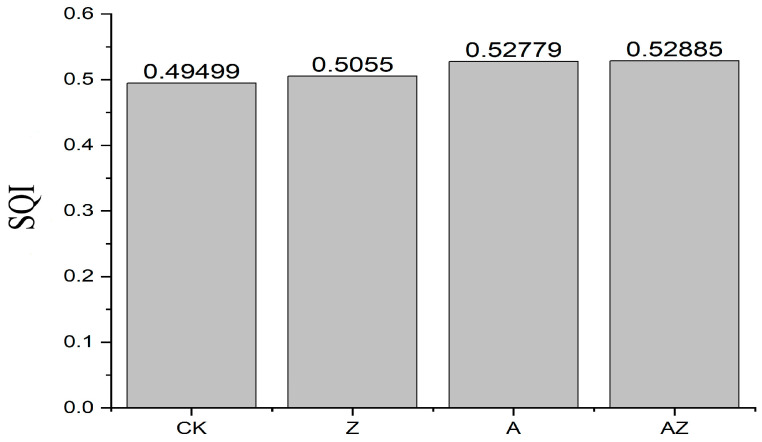
Soil comprehensive quality index of different treatments.

**Table 1 microorganisms-13-01778-t001:** The physical and chemical properties of soil under different treatments.

Index	CK	Z	A	AZ
pH	8.11 ± 0.13a	8.15 ± 0.13a	8.18 ± 0.13a	8.27 ± 0.17a
EC/(μs·cm^−1^)	59.93 ± 5.64b	55.3 ± 7.07b	66.53 ± 3.46b	87.9 ± 13.48a
SWC/%	9.3 ± 1.1bc	12.4 ± 0.7a	9.1 ± 0.6c	11.3 ± 0.9ab
SOC/(g·kg^−1^)	7.09 ± 0.72c	8.09 ± 0.33bc	9.44 ± 0.55b	15.23 ± 1.36a
TN/(g·kg^−1^)	0.26 ± 0.01c	0.41 ± 0.01b	0.43 ± 0.03b	0.47 ± 0.03a
TP/(g·kg^−1^)	0.13 ± 0.01c	0.34 ± 0.01a	0.31 ± 0.01b	0.34 ± 0.01a
C/N	27.45 ± 3.68b	19.96 ± 1.4c	22.19 ± 1.84c	32.09 ± 1.09a
C/P	54.7 ± 9.33a	24 ± 1.39b	30.02 ± 1.11b	45.28 ± 5.3a
N/P	1.36 ± 0.07bc	1.41 ± 0.12b	1.99 ± 0.13a	1.2 ± 0.03c
AN/(mg·kg^−1^)	102.67 ± 10.69b	149.33 ± 14.57a	172.67 ± 10.69a	184.33 ± 39.8a
AP/(mg·kg^−1^)	15.77 ± 0.6d	38.68 ± 0.85a	28.28 ± 1.86c	31.56 ± 1.75b
AK/(mg·kg^−1^)	20.4 ± 1.9c	21.27 ± 0.95c	31 ± 0.36b	38.03 ± 0.72a
EPS/(mg·kg^−1^)	0.0485 ± 0.0004d	0.8339 ± 0.0021b	0.6837 ± 0.0017c	0.9131 ± 0.0022a

Note: The statistical differences within a column are indicated by different letters (one-way ANOVA, α = 0.05).

**Table 2 microorganisms-13-01778-t002:** Stability of soil water-stable aggregates under different treatments.

Treatments	MWD (mm)	GMD (mm)	R0.25 (%)	PAD (%)	E_LT_ (%)	K
CK	0.61 ± 0.01c	0.44 ± 0.02b	0.48 ± 0.01b	0.43 ± 0.01a	0.52 ± 0.01a	0.21 ± 0.01a
Z	0.74 ± 0.01b	0.49 ± 0.04ab	0.5 ± 0.01b	0.42 ± 0.02a	0.5 ± 0.01a	0.2 ± 0.01ab
A	0.74 ± 0.01b	0.5 ± 0.02ab	0.53 ± 0.06b	0.39 ± 0.07ab	0.47 ± 0.06a	0.2 ± 0.01ab
AZ	0.99 ± 0.03a	0.64 ± 0.12a	0.65 ± 0.01a	0.29 ± 0.01b	0.35 ± 0.01b	0.16 ± 0.02b

Note: statistical differences within a column are indicated by different letters (one-way ANOVA, α = 0.05). MWD: mean weight diameter, GMD: geometric mean diameter, E_LT_: instable aggregate index, K: soil erodibility factor, R0.25: mass percentage of water-stable aggregates (>0.25 mm), PAD: percentage of aggregate destruction.

**Table 3 microorganisms-13-01778-t003:** Physicochemical indexes of *Poa annua* under different treatments.

Treatments	Total Length (cm)	Plant Height (cm)	Root Length (cm)	Total Weight (g)	Above-Ground Biomass (g)	Underground Biomass (g)	LNC (g/kg)	LPC (g/kg)
CK	8.5 ± 0.9j	4.5 ± 0.2h	4 ± 0.9g	1.66 ± 0.25i	0.35 ± 0.03c	1.31 ± 0.28i	20.3 ± 0.35i	0.23 ± 0.01k
Z	16.6 ± 1i	8.4 ± 0.4g	8.2 ± 0.7ef	4.64 ± 1.33def	0.71 ± 0.74bc	3.93 ± 0.6efg	25.67 ± 0.44g	1.18 ± 0.17cdefg
A	20.8 ± 0.8def	10.4 ± 0.3cde	10.4 ± 0.5abc	5.66 ± 0.6cde	0.53 ± 0.02bc	5.13 ± 0.58de	38.7 ± 0.66c	1.61 ± 0.04ab
AZ	24.2 ± 2.5b	12.6 ± 1.2b	11.6 ± 1.4a	13.15 ± 1.05b	1.15 ± 0.09ab	12 ± 1.06c	43.25 ± 0.74a	1.74 ± 0.03a

Note: statistical differences within a column are indicated by different letters (one-way ANOVA, α = 0.05).

**Table 4 microorganisms-13-01778-t004:** Results of soil properties from principal component analysis and weight values of soil properties.

Index	Principal Component 1	Principal Component 2	Principal Component 3	Weight
	Loading Coefficient	Loading Coefficient	Loading Coefficient	
pH	0.973	0.229	0.034	0.0534
EC	0.82	0.573	0.01	0.0554
SOC	0.922	0.369	0.121	0.0562
AN	0.952	−0.189	−0.241	0.0544
AP	0.62	−0.744	0.25	0.0543
TN	0.954	−0.285	−0.092	0.0549
TP	0.83	−0.557	−0.023	0.0556
SWC	0.328	−0.425	0.844	0.0404
C/N	0.355	0.9	0.253	0.0461
C/P	−0.261	0.948	0.181	0.0419
N/P	−0.668	0.743	0.029	0.0532
AK	0.905	0.347	−0.245	0.0567
EPS	0.877	−0.472	0.088	0.0565
MBC	0.986	0.164	−0.024	0.0520
MBN	0.97	0.131	−0.206	0.0531
MBC/MBN	−0.908	0.035	0.417	0.0506
MWD	0.966	0.163	0.202	0.0537
GMD	0.947	0.26	0.189	0.0554
R0.25	0.92	0.376	0.111	0.0562
Characteristic Root	13.107	4.563	1.329	/
Variance Contribution Rate %	68.99	24.02	7.00	/
Cumulative Variance Contribution Rate %	68.987	93.004	100	/

Note: “/” indicates that there is no value.

## Data Availability

The original contributions presented in this study are included in the article/Appendix A. Further inquiries can be directed to the corresponding authors.

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
