# Peer review of "Co-Inoculation Between Bacteria and Algae from Biological Soil Crusts and Their Effects on the Growth of *Poa annua* and Sandy Soils Quality"

_microorganisms, 2025, doi:10.3390/microorganisms13081778_

Round 1

Reviewer 1 Report

Comments and Suggestions for Authors

In this study, the authors investigated how the addition of a microbiological fertilizer composed of bacteria and algae affects the quality of biological soil crusts and the growth of bluegrass. The study is conceptually interesting; however, it lacks sufficient microbiological detail that would better explain the role of microorganisms in soil enrichment and plant growth promotion. In addition, the manuscript contains several issues that need to be addressed.

Title: Instead of BSC, write the full term.

Abstract: Line 17 – Rewrite provide protection for vegetation growth

Materials and Methods:

Section: Test materials – Separate the description of soil and microorganisms.

Explain the source of microbial strains (i.e., where the bacteria and algae were isolated from).

Describe the identification process at the species level.

Specify the culture media and conditions used to grow these microorganisms before inoculum preparation (e.g., media, temperature, light conditions, and duration of cultivation).

Line 76 – “Four strains of functional bacteria”

Clarify the term functional bacteria. What does it mean in this context? Were the bacterial strains characterized for specific traits that may have beneficial effects on soil quality?

It would strengthen the microbiological foundation of the work to include microbiological traits of the used strains (e.g., nitrogen fixation, phosphate solubilization, phytohormone production).

Line 80 –“The four green algae with the fastest growth rate were also selected”

This sentence is unclear. How was the growth rate determined? Among which strains was the selection made?

Provide the method of growth rate measurement (e.g., optical density, dry weight, chlorophyll content). Show growth curves of algal strains to justify selection

Line 89 – Add the scientific name of the plant species and seed producer

Control Groups

Why were there no control treatments containing only the A, Z, or AZ solutions?

Without these, it's unclear whether the observed changes in the soil were caused by the interaction of plants and microorganisms or by the solutions themselves.

You should include such controls to accurately interpret results.

Sections Soil Physicochemical Analysis, Determination of Microbial Biomass

Determination of Plant Growth and Physicochemical Indexes  – All methods used must be clearly described with:

  • The protocol or literature reference followed
  • The name and model of instruments used (with manufacturer and country of origin)

Results:

Lines 140–151 – Statistical significance

Table 1 shows no statistically significant differences in pH between treatment groups.

Therefore, all interpretations of the results in Lines 149–172 should be based solely on statistically significant findings as presented in Table 1.

Lines 184–192 – Define all abbreviations (MWD, GMD, R0.25, PAD, WLT i K) used in the text and below Table 2:

Line 201 – Define abbreviations MBC and MBN

Line 215 – Define abbreviations SOC, TN, AN, AK and EPS in the text and below Figure 3.

Below each figure and table that includes statistical comparisons, indicate which statistical test was used and clearly describe how statistical significance is shown (e.g., different letters, asterisks)

Discussion:

Line 325 – Replace the word “scholars” with “researchers”

Author Response

Thank you for pointing these out. We agree with these comments. Therefore, we have changed BSCs (Line 2-3) to Biological Soil Crusts in the “Title”, and rewritten "provide protection for vegetation growth" (Line 22) in the “Abstract”. In the “Materials and Methods”, we separately described the sources of soil and microorganisms, and added Sections 2.2 (Line 90-115) and 2.3 (Line 116-130) to explain the sources of microorganisms, where Section 2.3 (Line 116-130) explains how algae were selected. Supplementary Figure S7 provides the growth curves of the used algae strains. We also added Section 2.4 (Line 131-140) to explain the culture medium and conditions for culturing microorganisms. Supplementary Table S1 describes the bacterial species and Table S2 describes the functions of the selected bacteria. We changed the names of the used plants to Latin names (Line 145), and added the source of the seeds (Line 151-152). The methods for physical and chemical analysis of soil, plants, and microorganisms, as well as the names and models of the instruments used, are described in Supplementary Note S1 (Line 154), Note S2 (Line 159), and Section 2.8 (Line 162, Line 166) of the main text. Although the pH values did not show significant differences, there was an improvement, so the description was retained (Line 210-212). We defined all abbreviations used in Section 3.2 (Line 250-254, Line 269-270), Section 3.3 (Line 274, Line 287), and Section 3.4 (Line 293-296, Line 303) of the main text. Below each chart and table containing statistical comparisons, the methods of statistical tests and the display of statistical significance are added (Line 237, Line 265, Line 268, Line 286, Line 318, Line 359). We replaced "scholars" with "researchers" (Line 410).

Reviewer 2 Report

Comments and Suggestions for Authors

Comments to manuscript “Co-Inoculation Between Bacteria and Algae from BSCs and Their Effects on Plant Growth and Soil Quality”, submitted by Peng et al.

Peng et al. investigated the physicochemical properties of sandy soil samples and the beneficial influence of microorganisms on the soil properties. The four bacteria used for the study were isolated from biological crusts and identified as Paenibacillus mucilaginosus, Bacillus aryabhattai, Acinetobacter oleivorans and Arthrobacter bambusae (combined as bacteria solution A). Four “green algae” Auxenochlorella protothecoides, Coelastrella thermophila, Scenedesmus vacuolatus and Scenedesmus vacuolatus (combined as algae solution Z) were from the same crusts. Bacteria solution A and algae solution Z were also mixed (bacteria-algae solution AZ) and applied to soil samples, as solution A and Z, respectively. When soil samples were inoculated with solution A, solution Z and especially with the mixture AZ, the soil quality and the stability of soil aggregates were improved as verified by appropriate methods. The authors used a “bluegrass“for testing the effects of the inoculated soil on the growth (weight, aboveground and underground biomass, the total length, height, and root length, and the nitrogen and phosphorus contents. Peng et al. assumed that the growth and development of plants is improved by the treatment and concluded that such inoculations could be of practical value.

Major concerns

The authors wrote in the introduction:” However, there is still a lack of research on the effects of bacteria and algal interaction on plant growth and poor soil quality. Therefore, this study considered the application of microorganisms and algae in soil biological crust to plant growth…… The aim of this study is to provide theoretical basis and applicable technology for microbial application of barren land ecological restoration under the principle of economic practicality and ecological sustainability.”

In the study, there is no experiment for elucidation of bacteria and algal interactions nor if they interact with the plant. How can ecological restoration occur, when it is unclear whether the microorganisms applied in large amounts can survive? The sandy soil with poor fertility lacks nutrients. It is not surprising that nutrients were increased by the inoculation with the microorganisms (The bacteria-algal solution was applied once every 7 days, and the substrate was sprayed with water every day for the rest of the time to keep wet. The culture cycle was 90 days.) However, stay the microorganisms alive? Do they propagate? No experiment was performed that demonstrates the survival rate of the applied species. To me, the application of the microorganisms seems more to be a special way of fertilization and the increasing amounts of nutrients are originated from the decomposition of dead microorganisms, perhaps of all, some, or only of one applied species. The methods used for microbial biomass determination cannot give information. The demonstration of microbial survival/reproduction is essential.

The title of the manuscript includes effects on plant growth. Only one species, a monocot, was tested. The scientific name of the species is not given. Why was the choice” bluegrass”? It is necessary to study different species, also dicots. It has to be expected that different plant species will differently interact with the living microorganisms. Studies to elucidate interactions of even one of the microbial species with the bluegrass are missing. The work does not provide any insight in a plant-microbe or plant-bacteria-algae interaction. The discussion remains speculative.

A recommendation: Perhaps, the authors should restrict the manuscript on the soil parameter.

Minor concerns

There are numerous type and formatting errors in the text.

Line 48 and the following lines: the plural of genus is genera

Author Response

Comments 1: 
In the study, there is no experiment for elucidation of bacteria and algal interactions nor if they interact with the plant. How can ecological restoration occur, when it is unclear whether the microorganisms applied in large amounts can survive? The sandy soil with poor fertility lacks nutrients. It is not surprising that nutrients were increased by the inoculation with the microorganisms (The bacteria-algal solution was applied once every 7 days, and the substrate was sprayed with water every day for the rest of the time to keep wet. The culture cycle was 90 days.) However, stay the microorganisms alive? Do they propagate? No experiment was performed that demonstrates the survival rate of the applied species. To me, the application of the microorganisms seems more to be a special way of fertilization and the increasing amounts of nutrients are originated from the decomposition of dead microorganisms, perhaps of all, some, or only of one applied species. The methods used for microbial biomass determination cannot give information. The demonstration of microbial survival/reproduction is essential.

Response 1: 
Thank you for pointing these out. This article discusses the impact of the combined inoculation of bacteria and algae on Poa annua and barren soil, rather than the specific interactions between bacteria and algae. Therefore, the interaction is not covered in this article. We have only reached the stage of indoor control experiments at present. There is still a long way to go to achieve practical application, which is beyond the scope of this research. This experiment focuses only on whether the mixture of bacterial and algal solutions has a positive effect on plants and soil. After excluding the influence of nutrients in the culture medium on them, it can also be understood as a special form of fertilization. 

Comments 2: 
The title of the manuscript includes effects on plant growth. Only one species, a monocot, was tested. The scientific name of the species is not given. Why was the choice” bluegrass”? It is necessary to study different species, also dicots. It has to be expected that different plant species will differently interact with the living microorganisms. Studies to elucidate interactions of even one of the microbial species with the bluegrass are missing. The work does not provide any insight in a plant-microbe or plant-bacteria-algae interaction. The discussion remains speculative.

Response 2:
We narrowed down the definition of “plants” in the title(Line3) and provided the scientific name. We also explained why we chose Poa annua. (lines 69-71). This experiment only discusses the effects on Poa annua. Your suggestions are very inspiring to us. In the future, we will study more different species to improve the relevant theories. Because there is currently a lack of research on microorganisms and Poa annua, we conducted this experiment.

Comments 3: 
There are numerous type and formatting errors in the text.
Line 48 and the following lines: the plural of genus is genera

Response 3:
We have improved the writing of the article, correcting the type and formatting errors in the text.

Reviewer 3 Report

Comments and Suggestions for Authors

The paper of Peng et al., titled “Co-Inoculation Between Bacteria and Algae From BSCs and Their Effects on Plant Growth and Soil Quality” is an incomplete contribution. Though some of the data might be interesting, the way the experiments are described do not meet the expectations of a serious paper. The authors provide supplementary materials, but these are not cited in the text. Also, in the main MS the legend of the supplementary materials does not correspond to what was provided. Overall, the writing is not good, a serious revision of the text with the help of somebody who has a better comment of English language is needed.

Most of the references are not properly formatted and the citation in the MS is not OK.

For example, in line 28 it is mentioned “Zhang Jianfeng et al.”, no year, but citation [1] is the paper of Rattan L.

Same, in line 34, the authors mention “ Zhao Yitai et al”, and again, no year; at citation [2] it is mentioned the paper of Guo Xiaona et al! This is an example of academic integrity violation because improper citation. How can such a work be trusted?

Overall, it is clear that the authors were supposed to spend more time working on the MS before having it submitted!

The current MS is not possible to be evaluated as it is. Also, the results are not supported by the experimental design.

The authors have to indicate clearly how the algal mix and the bacterial mix were produced, if the solution that was applied to plants contained some of the medium on which the microorganisms were grown, and so on. All this information is missing. There were 12 applications of these mixes, so one have to understand if the minerals in the medium contributed to the observed growth or not. If they were added with the medium, then controls had to be represented by the media as well, not only by water. There is a very large difference reported in table 3 which, without good control data it is hard to understand these results. Also 100 plants growing on a pot with a diameter of 8 cm gives approximately 2 plants per square centimeter, which would be the regular amount seeded to get a nice, healthy lawn, on a nutrient rich soil. It is very unlikely that a sandy soil can sustain that type of growth, so this explains the number of treatments and the amount applied each time, which is quite large! At this point it is not obvious why the experiment was conducted, as at this input, the outcome should be obvious. Anyhow, the authors have to clearly describe the composition of the media in which the microorganisms were grown and if such media was applied along with the mix of microorganisms!

 Also, the authors should not include information which correspond to general knowledge such as in Figure 3S where it is indicated that “Marker bands can be used as the standard to roughly estimate the molecular weight of DNA samples.”

Comments on the Quality of English Language

English language needs to be improved. 

Author Response

Comments 1: 
The paper of Peng et al., titled “Co-Inoculation Between Bacteria and Algae From BSCs and Their Effects on Plant Growth and Soil Quality” is an incomplete contribution. Though some of the data might be interesting, the way the experiments are described do not meet the expectations of a serious paper. The authors provide supplementary materials, but these are not cited in the text. Also, in the main MS the legend of the supplementary materials does not correspond to what was provided. Overall, the writing is not good, a serious revision of the text with the help of somebody who has a better comment of English language is needed.
Most of the references are not properly formatted and the citation in the MS is not OK.
For example, in line 28 it is mentioned “Zhang Jianfeng et al.”, no year, but citation [1] is the paper of Rattan L.Same, in line 34, the authors mention “ Zhao Yitai et al”, and again, no year; at citation [2] it is mentioned the paper of Guo Xiaona et al! This is an example of academic integrity violation because improper citation. How can such a work be trusted?Overall, it is clear that the authors were supposed to spend more time working on the MS before having it submitted!
Also, the authors should not include information which correspond to general knowledge such as in Figure 3S where it is indicated that “Marker bands can be used as the standard to roughly estimate the molecular weight of DNA samples.”

Response 1: 
Thank you for pointing this out. We are deeply sorry. Before submitting the manuscript, we failed to carefully review the references, which was a major mistake and fault on our part. We have double-checked and revised each piece of literature in accordance with the required format. Furthermore, we have also revised and improved the supplementary materials and incorporated them in the text.

Comments 2: 
The authors have to indicate clearly how the algal mix and the bacterial mix were produced, if the solution that was applied to plants contained some of the medium on which the microorganisms were grown, and so on. All this information is missing. There were 12 applications of these mixes, so one have to understand if the minerals in the medium contributed to the observed growth or not. If they were added with the medium, then controls had to be represented by the media as well, not only by water. There is a very large difference reported in table 3 which, without good control data it is hard to understand these results. Also 100 plants growing on a pot with a diameter of 8 cm gives approximately 2 plants per square centimeter, which would be the regular amount seeded to get a nice, healthy lawn, on a nutrient rich soil. It is very unlikely that a sandy soil can sustain that type of growth, so this explains the number of treatments and the amount applied each time, which is quite large! At this point it is not obvious why the experiment was conducted, as at this input, the outcome should be obvious. Anyhow, the authors have to clearly describe the composition of the media in which the microorganisms were grown and if such media was applied along with the mix of microorganisms!

Response 2:
We added sections 2.2-2.4 to explain how the algal mixture and bacterial mixture were produced, and to clarify the types of solutions used for plants (Line 90-140). Supplementary materials Table S3, together with the description in the first paragraph of Section 3.1 of the manuscript (Line 202-209), jointly explains the influence of the culture medium on the results and the selection of control groups in the pot experiments. Based on the results of Poa annua 's pre-germination experiments in the laboratory, the seeding amount used in this study was determined to be appropriate.

Round 2

Reviewer 1 Report

Comments and Suggestions for Authors

Dear Authors,
Your response was difficult to follow. In the future, please address reviewers’ comments using a clear point-by-point format rather than a general summary, as this improves clarity and facilitates evaluation. Additionally, some of my questions remained unanswered. However, given that the microbiological section has been substantially improved, I will not insist on further clarification at this time.

Author Response

Thank you very much for your suggestions !

Reviewer 2 Report

Comments and Suggestions for Authors

The manuscript was improved in a way that may allow acceptance.

Author Response

Thank you very much for your suggestions!

Reviewer 3 Report

Comments and Suggestions for Authors

The revised manuscript by Peng et al., titled “Co-Inoculation Between Bacteria and Algae From BSCs and Their Effects on Plant Growth and Soil Quality,” remains an incomplete contribution. As noted in the previous review, the quality of the writing is substandard and requires substantial revision, ideally with the assistance of a fluent English speaker or professional editor. Additionally, formatting issues persist, particularly with the improper citation of references—an issue that was already highlighted in the earlier review. It is unclear why these fundamental concerns were not addressed prior to resubmission, raising questions about the manuscript’s readiness for publication.

For example, in line 34, it is mentioned Zhang Jianfeng et al.(2018). A correct citation was supposed to be, Zhang et al. [1].

Same in lines 40 ( Zhao Yitai), 53 (Zhou Zhigang), and so on. The authors should familiarize themselves with how to properly cite references.

Introduction

The introduction lacks conciseness and fails to highlight the most significant international findings in the field. As a result, it does not effectively establish the relevance or potential impact of the current research. The final paragraph is particularly problematic, as it presents a disorganized mix of background information, preliminary results, and unclear writing. Additionally, the scientific name Poa annua should be accompanied by its common name, bluegrass, for clarity. Furthermore, the phrasing “treatments inoculated onto Poa annua” is imprecise and should be revised for accuracy.

Materials and methods

Significant information has been added but most of it is not well structured and has to be written in a more concise way. Conflicting information is provided in several instances. For example, in line 81 we learn that : “the soil was obtained from Daxing District of Beijing”. However, in supplementary materials, Figure 1 it is mentioned that: “The biological crusts were collected from the karst desertification area in Yunnan Province.” It is very hard to understand what was done.

Also, information from materials and methods section should not be mixed with those in the results section. Significant re-writing is needed because of poor usage of English language. An example of mixed methods and results and poor writing can be found in lines 94-97 “After single colonies grew, the samples were transferred to 10% TSB liquid medium and cultured in a 30 ℃, 180 rpm shaking incubator for 24 to 48 hours, and 862 strains were screened (Supporting Information: Figure S2)”

Example of incorrect writing: “in a 30 ℃, 180 rpm shaking incubator”

Figure S2 from the supporting information represents results.

Lines 97-113 includes a listing of methods that are not detailed, with no references, and results. This is another example of poor structuring of the paper.

“Whereafter, functional screening was conducted, including inorganic phosphorus removal function, potassium removal function, nitrogen fixation function, iron-producing carrier function, IAA production function, EPS production 99 function and ACC deaminase production function. As a result, 162 strains were further selected for further studies (Supporting Information: Figure S3-S4). Based on the data of the commonly mentioned beneficial bacteria genera in the literature, the pathogenic genera could be excluded, and a certain degree of classification redundancy could be avoided. Finally, 36 strains of bacteria were obtained (Supporting Information: Table S1,S2).

Results section

The results section should be re-written in a more concise manner and incorporate the data from materials and methods.

It is hard to believe that the authors start presenting the results with “During the pot experiments in section 2.2” Actually, the pot experiments are presented in subsection 2.5. Section 2.2 is “Isolation of Functional bacteria”!

It is unclear why the authors have not explicitly identified the four functional bacterial and algal strains used in the study. Without this fundamental information, it is difficult to evaluate the scientific merit and reproducibility of the work. Clear reporting of the microbial strains is essential for assessing the novelty and applicability of the findings.

Discussion and conclusions

The discussion and conclusions presented are overly general. While the use of a combination of bacteria and algae to enhance soil quality is indeed beneficial, this concept is well-established and supported by research conducted decades ago. To strengthen the manuscript, the authors should clarify the uniqueness of their microbial consortium. Specifically, they should identify which bacterial and algal species were used, and explain how the known properties of these microorganisms contributed to the observed outcomes. This would help distinguish their work from prior studies and highlight its scientific contribution.

Comments on the Quality of English Language

Significant improvement of English language usage is needed.

Author Response

Comments 1: 
The revised manuscript by Peng et al., titled “Co-Inoculation Between Bacteria and Algae From BSCs and Their Effects on Plant Growth and Soil Quality,” remains an incomplete contribution. As noted in the previous review, the quality of the writing is substandard and requires substantial revision, ideally with the assistance of a fluent English speaker or professional editor. Additionally, formatting issues persist, particularly with the improper citation of references—an issue that was already highlighted in the earlier review. It is unclear why these fundamental concerns were not addressed prior to resubmission, raising questions about the manuscript’s readiness for publication.

Response 1: 
Thank you for pointing these out.The citation style of the references in the current manuscript introduction was modified according to the suggestion of the first editor. There might be differences of opinion between you two. However, we have made the modifications based on your suggestions(Line36、42、54). During the rework phase, we have engaged professionals to make revisions and improvements, and they have been able to provide a proof of the revisions.

Comments 2: 
The introduction lacks conciseness and fails to highlight the most significant international findings in the field. As a result, it does not effectively establish the relevance or potential impact of the current research. The final paragraph is particularly problematic, as it presents a disorganized mix of background information, preliminary results, and unclear writing. Additionally, the scientific name Poa annua should be accompanied by its common name, bluegrass, for clarity. Furthermore, the phrasing “treatments inoculated onto Poa annua” is imprecise and should be revised for accuracy.

Response 2:
In the initial submission of our manuscript, we used common names. The first editor's suggestion was to use Latin names, which would be more formal. Therefore, we all changed them to Latin names. The phrasing “treatments” does not refer to the medical treatment meaning, but rather different liquid groups in the experiment. This is a common expression found in the literature. Furthermore, we made revisions to the introduction according to your suggestions(Line 68-80).

Comments 3: 
Significant information has been added but most of it is not well structured and has to be written in a more concise way. Conflicting information is provided in several instances. For example, in line 81 we learn that : “the soil was obtained from Daxing District of Beijing”. However, in supplementary materials, Figure 1 it is mentioned that: “The biological crusts were collected from the karst desertification area in Yunnan Province.” It is very hard to understand what was done.

Response 3:
You might have confused some information. The karst desertification areas in Yunnan mentioned in the supplementary materials are the source areas of the crusts. We isolated bacteria and algae from the crusts. While the barren soil in Daxing District, Beijing was used as the material for the pot experiment. These two are completely different experiments. So there is no conflict.

Comments 4: 
Also, information from materials and methods section should not be mixed with those in the results section. Significant re-writing is needed because of poor usage of English language. An example of mixed methods and results and poor writing can be found in lines 94-97 “After single colonies grew, the samples were transferred to 10% TSB liquid medium and cultured in a 30 ℃, 180 rpm shaking incubator for 24 to 48 hours, and 862 strains were screened (Supporting Information: Figure S2)”.
“Whereafter, functional screening was conducted, including inorganic phosphorus removal function, potassium removal function, nitrogen fixation function, iron-producing carrier function, IAA production function, EPS production 99 function and ACC deaminase production function. As a result, 162 strains were further selected for further studies (Supporting Information: Figure S3-S4). Based on the data of the commonly mentioned beneficial bacteria genera in the literature, the pathogenic genera could be excluded, and a
certain degree of classification redundancy could be avoided. Finally, 36 strains of bacteria were obtained (Supporting Information: Table S1,S2).

Response 4:
We have detailed the separation and screening processes of Sections 2.2 and 2.3 in Supplementary Materials Note S1 and Note S2 respectively, and presented the specific information of the obtained required bacterial and algal strains together(Supporting Information: Table S1, S8, S9).

Comments 5: 
The results section should be re-written in a more concise manner and incorporate the data from materials and methods.
It is hard to believe that the authors start presenting the results with “During the pot experiments in section 2.2” Actually, the pot experiments are presented in subsection 2.5. Section 2.2 is “Isolation of Functional bacteria”!

Response 5:
We have re-written the results section concisely according to your suggestions. You may not have fully understood the experimental methods in Section 2.2 . As the process of screening and identifying the final four bacterial strains was extensive and complex, and because strain screening served as a methodological tool rather than the primary focus of this study, these details were placed in the Supplementary Materials. Pot experiments were employed during the screening phase, as indicated in Supplementary information Note S1. Data from these pot experiments are presented in Supplementary Tables S3-S5. The detailed information of the 4 bacterial and algal strains used in this study is provided in Supplementary Tables: S1, S8, S9. 

Comments 6: 
The discussion and conclusions presented are overly general. While the use of a combination of bacteria and algae to enhance soil quality is indeed beneficial, this concept is well-established and supported by research conducted decades ago. To strengthen the manuscript, the authors should clarify the uniqueness of their microbial consortium. Specifically, they should identify which bacterial and algal species were used, and explain how the known properties of these microorganisms contributed to the observed outcomes. This would help distinguish their work from prior studies and highlight its scientific contribution.

Response 6:
Following your suggestion, we have rewritten the discussion and conclusion sections.